# A tool for identifying green solvents for printed electronics

Christian Larsen [1,2,3], Petter Lundberg [1,3], Shi Tang [1,2,3], Joan Ràfols-Ribé [1], Andreas Sandström [1,2], E. Mattias Lindh [1], Jia Wang [1] & Ludvig Edman [1,2✉]

The emerging field of printed electronics uses large amounts of printing and coating solvents during fabrication, which commonly are deposited and evaporated within spaces available to workers. It is in this context unfortunate that many of the currently employed solvents are non-desirable from health, safety, or environmental perspectives. Here, we address this issue through the development of a tool for the straightforward identification of functional and "green" replacement solvents. In short, the tool organizes a large set of solvents according to their Hansen solubility parameters, ink properties, and sustainability descriptors, and through systematic iteration delivers suggestions for green alternative solvents with similar dissolution capacity as the current non-sustainable solvent. We exemplify the merit of the tool in a case study on a multi-solute ink for high-performance light-emitting electrochemical cells, where a non-desired solvent was successfully replaced by two benign alternatives. The green-solvent selection tool is freely available at: www.opeg-umu.se/green-solvent-tool.

[1] The Organic Photonics and Electronics Group, Department of Physics, Umeå University, Umeå, Sweden. [2] LunaLEC AB, Umeå, Sweden. [3] These authors contributed equally: Christian Larsen, Petter Lundberg, and Shi Tang. ✉email: ludvig.edman@umu.se

Printed electronics enables for cost-efficient solution-based fabrication of functional and novel electronic and photonic devices, and, as such, it promises to develop into a multi-billion industry in the near future[1–6]. Large research and development efforts within academia and industry are currently dedicated to the design and development of improved printing and coating inks, because the functionality and performance of the resulting printed-electronic devices are strongly dependent on a variety of ink properties, such as solute solubility, viscosity, surface wettability, film-forming capacity, vapor pressure and shelf life[7–19].

However, an up-to-now often overlooked issue within the printed-electronics field is related to the sustainability of the employed solvents. Many of the currently employed solvents are not desirable for scale-up and commercial introduction since they present serious issues for health, safety, and/or environmental reasons. This is particularly problematic since it is anticipated that the fabrication of printed electronic devices, i.e. the deposition of the inks and the evaporation of the constituent solvents, often will be executed in open environments where workers will be exposed to the solvent vapors. Thus, it is easy to motivate why alternative "green" solvents, which deliver an ink performance on par with currently used non-sustainable solvents, should be identified[20–24].

In order to support and facilitate this transition to more sustainable solvents, we have developed a free online tool, which allows the user to identify greener functional replacement solvents for his/her particular application in a straightforward manner. More specifically, we first present the rationale for the organization and ranking of the solvent functionality with the aid of the Hansen solubility parameters and a number of key ink properties, then introduce the sustainability descriptors and ranking, and finally exemplify the utilization of the tool in a case study where two functional green replacement solvents for the solution-based fabrication of high-performance light-emitting electrochemical cells are identified.

## Results

**Green solvent definition.** So, what is the definition of a green solvent? This is not a trivial question to answer in a succinct manner, but informed parties can most probably agree that a qualified green solvent should exhibit a combination of low health hazard, high safety, and small environmental impact during its entire life cycle. The "Globally Harmonized System of Classification and Labelling of Chemicals" (GHS) provides qualitative assessments as regards to the health, safety, and environmental impact for a large number of solvents (and other chemicals) via its "Hazard and Precautionary Statements". A number of organizations, e.g., the American Chemical Society Green Chemistry Institute Pharmaceutical Roundtable and IMI: CHEM21, and companies, such as Pfizer, Sanofi, and GlaxoSmithKline (GSK), have utilized this and similar information for the quantitative classification and ranking of

the sustainability of a wide range of chemical compounds, including solvents[25–34]. Although the employed approaches differ slightly, e.g., in that the properties are weighted differently in the overall sustainability classification, it is reassuring that recent reviews have concluded that the classification and overall ranking were rather invariant between the different studies[35,36].

In this study, we have selected to employ the regularly updated GSK solvent sustainability guide, which quantitatively evaluates solvents according to a broad variety of different aspects[25–27]. Specifically, the guide grades the solvents in ten different subcategories, which are translated into four category scores, and finally summarized into a composite score value ($G$), using the procedure outlined in Table 1. All scores range between 1 and 10, with a low score implying nonsustainable properties in that specific category while a high score is desirable. It should be mentioned that if the solvent is inadequately characterized in a subcategory then GSK lowers the corresponding score. The GSK guide is attractive in that it contributes with both specific information on issues in the different subcategories (which might be more or less important in different environments and applications) and a facile ranking of the different solvents via the composite score, with the latter allowing the end-user to make a quick and informed decision on the solvent of choice.

**Functional solvent selection.** A functional solvent must obviously be capable of dissolving the solute(s), i.e., the solid material(s), in a desired concentration, and the rigorous definition is that a solvent can dissolve a solute if the total Gibbs free energy of the solvent-solute system is lowered during the dissolution process. However, an evaluation of whether this criterion is fulfilled requires specific measurements and calculations for each solvent-solute system.

A more practical and generic evaluation of the dissolution properties of a larger group of solvents is instead provided by the so-called Hansen method, which is inspired by the well-known "like-dissolves-like" approach[37]. The Hansen method separates the total cohesion energy (per molar volume) of a solvent into three Hansen solubility parameters (HSPs): (i) the dispersion energy that considers molecular dispersion interactions ($\delta_D$); (ii) the polar energy that considers molecular dipolar interactions ($\delta_P$); and (iii) the hydrogen-bonding energy that considers molecular hydrogen bonding interactions ($\delta_H$). The similarity in solubility capacity of two solvents is provided by their proximity in the 3D Hansen solubility space, with their effective separation ($R_a$) being calculated with the following equation:

$$R_a^2 = 4\left(\delta_{D1} - \delta_{D2}\right)^2 + \left(\delta_{P1} - \delta_{P2}\right)^2 + \left(\delta_{H1} - \delta_{H2}\right)^2 \quad (1)$$

The factor four in front of the dispersion term signals the relative importance of the dispersion parameter, and its origin is discussed in detail in references[37,38].

**Table 1 The solvent sustainability categories and subcategories in the GSK solvent sustainability guide and the method for the calculation of the composite score.**

| Category | Subcategory | Category Score | Composite Score |
|---|---|---|---|
| Health | Health Hazard<br>Exposure Potential | $H = \sqrt{HH \times EP}$ | $G = \sqrt[4]{H \times S \times E \times W}$ |
| Safety | Flammability & Explosion<br>Reactivity & Stability | $S = \sqrt{F\&E \times R\&S}$ | |
| Environment | Air Impact<br>Aqueous Impact | $E = \sqrt{Air \times Aqua}$ | |
| Waste Disposal | Incineration<br>Recycling<br>Bio Treatment<br>Volatile Organic Compounds | $W = \sqrt[4]{I \times R \times BT \times VOC}$ | |

Figure 1 presents the location of 132 solvents in the 3D Hansen solubility space, with some of the more well-known solvents being identified by their chemical name. At this point, we recommend the reader to consult with the web tool at www.opeg-umu.se/green-solvent-tool, where the same graph is presented with higher detail and in addition can be rotated for clearer visualization. Moreover by clicking on a desired solvent in the Hansen space graph or in the solvent ranking table in the web tool, information is disclosed regarding its chemical structure, HSPs, CAS number, melting and boiling points, viscosity, surface tension, specific health, safety and environmental issues, and its scores in the different GSK sustainability categories (as identified in Table 1).

The composite sustainability score, $G$, is practical in that it provides a summary evaluation of the sustainability of a solvent. In order to directly visualize this overall sustainability, the sphere that represents a solvent in Fig. 1 (and in the web tool) is both size and traffic-light-color coded, according to the specification in the inset. In short, a large green circle represents a highly sustainable solvent ($G \geq 7$), an intermediate-sized yellow/orange circle corresponds to a solvent with a limited number of sustainability issues ($G = 5–6$), while a small red circle is concomitant with a solvent that should be avoided ($G \leq 4$). As an example, we note that ethylene glycol ($G = 8.1$) and n-butyl acetate ($G = 7.5$) are preferable green solvents from most aspects, whereas benzene ($G = 3.7$) and 1–4 dioxane ($G = 4.1$) should be avoided.

The closer two solvents are positioned in the 3D Hansen solubility space in Fig. 1, i.e., the smaller the $R_a$, the more similar is their solubility capacity (and cohesive energy). In other words, if we know that a certain solvent can dissolve a specific solute, the probability that a second solvent also can dissolve the same solute is increasing with decreasing $R_a$. The web tool provides a facile route to rank all of the solvents as regards to their $R_a$ value with respect to a known functional solvent. The procedure is a follows: (i) mark the "known functional solvent(s) of your solute" option in the upper left corner, (ii) select the functional solvent in the selection bar below, and (iii) click the update button. The outcome is that all solvents will be ranked from low to high $R_a$ with respect to the selected functional solvent. For instance, if we

select toluene as the functional solvent for a specific solute, we are quickly informed that cumene ($R_a = 0.8$) and benzene ($R_a = 1.6$) are likely to be functional solvents for the same solute, but not acetonitrile ($R_a = 17.9$) or ethylene glycol ($R_a = 25.9$).

A refinement of this solvent ranking procedure is available if two or more solvents are known to dissolve the desired solute(s). All of the identified functional solvents are in this scenario included into the selection bar in step (ii); after the update button is pushed, the program calculates the mean HSP values for these known functional solvents and ranks the remaining solvents as regards to their $R_a$ distance from this mean of the HSPs of the functional solvents. A further improvement of the solvent ranking is possible when the HSPs of the *solute* are known. Then the solvent-ranking procedure follows the path: (i) mark the "known HSP of your solute", (ii) include the values for $\delta_D$, $\delta_P$, and $\delta_H$, and (iii) click the update button.

The functionality of an ink solvent is further determined by its ability to enable a desired ink-substrate wetting, ink-film formation, and solute-film drying during the printing/coating procedure. Key solvent properties that determine if a specific solvent is suitable for a certain printing/coating process and substrate include its boiling point (which is related to the vapor pressure), viscosity and surface tension, and the web tool presents the tabulated values for each of these properties for all the solvents. It also allows the user to define the functional range of each of these properties for a specific application by clicking on the "Refinement options" tab and by dragging the corresponding sliders. By pressing the update button, the tool excludes solvents with properties outside the selected ranges, and also updates the solvent ranking table and the Hansen space plot accordingly. We note that for some solute combinations and applications, such as organic solar cells[24], it is common to utilize multi-solvent inks in order to attain the desired ink and dry-film properties. In this context, we mention that the tool is capable of identifying green replacements for each of the different solvents in a multi-solvent ink by simply repeating the above procedure for each solvent.

**Case study with green ink formulation for high-performance LEC device.** With this tool at hand, a viable and straightforward method to identify potential replacement solvents that are both functional (low $R_a$ value and appropriate physical properties) and sustainable (high $G$ value) is in place. We exemplify how such a solvent-replacement procedure can be executed in a case study on the identification of a sustainable solvent for the fabrication of a high-performance light-emitting electrochemical cell (LEC). A bright and record-efficient LEC, featuring a solution-processed single-layer active material sandwiched between two air-stable electrodes, was recently reported[39], but a drawback was that the active-material ink comprised chlorobenzene as the solvent. This is a concern since chlorobenzene is harmful to the skin (GHS hazard statements H312 and H315), the eyes (H319) and when inhaled (H332), as well as toxic to aquatic life (H411) and flammable (H226); see Table 2. Consequently, chlorobenzene scores low in both the health category ($H = 4.0$) and in the environmental category ($E = 3.7$), and the composite sustainability score is a modest $G = 5.4$. A further challenge with this LEC ink is that it comprises four different solutes (two host compounds, one guest emitter, and one electrolyte) that should be dissolved in its solvent in a high total solute concentration of ~30 g l$^{-1}$ (see the "Methods" section for details) in order to allow for the formation of a pinhole-free thin film following the ink deposition and drying.

Since chlorobenzene is an established functional solvent for this particular multi-component solute, we start by selecting chlorobenzene as the "known functional solvent(s)" in the selection bar in the web tool and by clicking the update button.

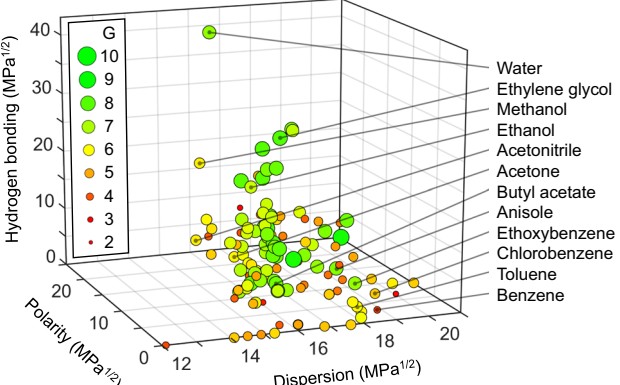

**Fig. 1 Visualization of the localization and the composite score, *G*, of the solvents in the Hansen space.** The distribution of 132 common solvents in the 3D Hansen solubility space, as spanned by the three Hansen solubility parameters: dispersion, polarity and hydrogen bonding. The effective $R_a$ distance between two solvents in the 3D Hansen space provides information on their similarity in solubility properties, with a smaller $R_a$ value indicating a higher similarity in solubility. The size and traffic-light-color coding of the sphere representing a particular solvent communicate its GSK composite score, as specified in the inset, with a large green sphere indicating a benign solvent while a small red sphere marks a problematic solvent.

**Table 2 Identification of green replacement solvents for LEC ink.**

| $R_a$ | Solvent (CAS number) | Dissolves LEC solutes[a] | Boiling point (°C) | Viscosity (mPa s) | Surface tension (mN m⁻¹) | GSK[25] | | | | | GHS[44,45] |
|---|---|---|---|---|---|---|---|---|---|---|---|
| | | | | | | Health | Safety | Environmental | Waste | Composite score | Hazard statements[b] |
| 0 | Chlorobenzene (108-90-7) | Yes | 132 | 0.81 | 33.5 | 4.0 | 8.9 | 3.7 | 6.5 | 5.4 | H226 H312 H315 H319 H332 H411 |
| 2.3 | Ethoxybenzene (103-73-1) | Yes | 170 | 9.76 | 32.7 | 4.9* | 9.5 | 6.7 | 8.7 | 7.2 | H226 |
| 5.5 | Anisole (100-66-3) | Yes | 154 | 1.52 | 34.2 | 7.5 | 7.9 | 6.5 | 8.0 | 7.4 | H226 H315 H319 |
| 5.7 | Cyclohexanone (108-94-1) | Yes | 155 | 2.20 | 35.1 | 6.5 | 8.5 | 6.9 | 7.2 | 7.2 | H226 H302 H312 H315 H318 H332 |
| 6.2 | Methyl oleate (112-62-9) | No | 218 | 4.88 | 31.3 | 6.3 | 8.4 | 6.3 | 8.1 | 7.2 | H319 |
| 7.2 | 2-Ethylhexyl acetate (103-09-3) | No | 199 | 1.30 | 27.5 | 8.4 | 9.5 | 4.9 | 8.7 | 7.6 | H315 |
| 7.7 | Pentyl acetate (628-63-7) | No | 149 | 0.92 | 24.7 | 7.5 | 8.9 | 6.7 | 7.7 | 7.7 | H226 |
| 7.7 | n-Butyl acetate (123-86-4) | No | 126 | 0.69 | 23.0 | 8.9 | 8.9 | 4.9 | 8.0 | 7.5 | H226 H336 |

Seven green ($G > 7$) potential replacement solvents are ranked from low to high Ra with the currently employed non-green ($G = 5.4$) chlorobenzene solvent included at $R_a = 0$. The table also presents the experimentally verified dissolution capacity, a number of relevant physical parameters, detailed category scores, the overall composite score, and the GHS hazard statements.
[a]at ≥30 g l⁻¹ concentration.
[b]H226: Flammable liquid and vapor; H312/H332: Harmful if in contact with skin/if inhaled; H315/H318/H319: Causes skin irritation/serious eye damage/serious eye irritation; H336: May cause drowsiness or dizziness; H411: Toxic to aquatic life with long-lasting effects.
*Lack of data result in a lowered subcategory score.

This updates the solvent ranking table in the web tool and sorts all 132 solvents from lowest-to-highest $R_a$. The result in the solvent ranking table can be refined to only include sustainable solvents by first clicking on the "Refinement options" tab, thereafter dragging the slider under "Set lower limit for G" until it shows $G > 7$, and by finally clicking the update button. Table 2 reveals that the seven closest sustainable solvents (with $G \geq 7$), ranked from lowest to highest $R_a$ (i.e. from highest to lowest probability of dissolution functionality), are: ethoxybenzene, anisole, cyclohexanone, methyl oleate, 2-ethylhexyl acetate, pentyl acetate, and n-butyl acetate.

We start our experimental investigation with the closest neighbor in the Hansen solubility space, ethoxybenzene ($R_a = 2.3$), and establish that it is capable of dissolving the multi-component solute in the desired concentration. Since ethoxybenzene features a significantly higher composite sustainability score ($G = 7.2$) than chlorobenzene ($G = 5.4$), it is clearly a more preferred solvent from a sustainability perspective. In fact, Table 2 discloses that ethoxybenzene scores higher than chlorobenzene in all four category scores. The second-nearest neighbor, anisole ($R_a = 5.5$), also passes the solubility test, and since it scores slightly higher in the composite sustainability score ($G = 7.4$) it could be an even better option. It should however be noted that ethoxybenzene scored a modest 4.9 in the health category because of inadequate information[25], and that it is thus likely that a complete evaluation would have resulted in a higher composite sustainability score. We mention that a third solvent, cyclohexanone, also passed the solubility test, but that it was not considered further since it did not present an improvement in the composite sustainability and since it is labeled with a larger number of hazard statements; see Table 2. The three acetates—2-ethylhexyl acetate, pentyl acetate, and n-butyl acetate—as well as methyl oleate are interesting from a sustainability viewpoint, but could not dissolve the multicomponent solute in the desired high concentration. This indicates that the $R_a$ boundary for dissolution for this particular multi-solute system has been crossed at $R_a > 6$, although it should be emphasized that this boundary is not expected to be distinct.

With ethoxybenzene and anisole identified as potential green replacement solvents to chlorobenzene for the LEC ink, we turn to device fabrication and characterization in order to investigate the practical functionality of the different inks in LEC devices. It was first established that the chlorobenzene, ethoxybenzene and anisole based inks all could be used for repeatable spin-coating fabrication of uniform and pinhole-free active-material thin films (with a thickness of 120 nm), as evidenced by the low surface roughness in AFM and the spatial uniformity of the UV-activated photoluminescence (see Fig. S1). Note that the differences in viscosity and boiling point for the ink solvents (see Table 2) were compensated for by the selections for solute concentration and spin speed, as detailed in the Methods section. We fabricated six indium-tin-oxide/poly(3,4-ethylenedioxythiophene):poly(styrene sulfonate)/active-material/Al devices with each fresh ink (prepared the same day), and Fig. 2a present the luminance and voltage transients as a function of ink-solvent selection for a typical device in each category when driven by a constant current density of $j = 77$ A m⁻².

The transients are highly independent on the ink-solvent selection, and all devices feature the LEC-characteristic increasing luminance and decreasing voltage during the turn-on phase when a p–n junction doping structure forms in the active material[40,41]. All three devices also emit vibrant green light ($\lambda_{peak} = 525$ nm) with essentially identical electroluminescence spectrum (see Figure S2), which implies that the guest emitter is molecularly dispersed in the host matrix and not forming emission-shifting aggregates. Table S1 presents a summary of key performance

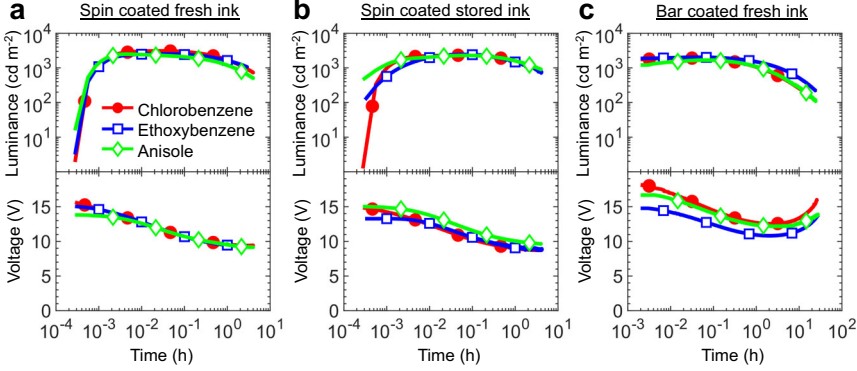

**Fig. 2 LEC performance as a function of ink solvent, ink storage and ink deposition method.** The temporal evolution of the luminance and the voltage for LECs fabricated from inks based on chlorobenzene (solid red circles), ethoxybenzene (open blue squares) and anisole (open green diamonds). **a** The performance of LECs fabricated by spin coating "fresh" inks (prepared the same day). **b** The performance of LECs fabricated by spin coating "old" inks, which had been stored for 30 days following ink preparation. **c** The performance of LEC fabricated by scalable bar coating. All devices were driven by a constant current density of $j = 77$ A m$^{-2}$.

metrics of the best-performing device in each category. It reveals that the turn-on time to a high luminance of >1000 cd m$^{-2}$ is very fast (<2 s) and effectively identical for all devices, that the peak luminance and efficiency is a bit higher for the LECs fabricated from the chlorobenzene ink (3100 vs. ~2540 cd m$^{-2}$ and 39.9 vs. ~32.9 cd A$^{-1}$), and that the operational lifetime is essentially the same for all devices.

We have also investigated the storage stability of the three inks based on chlorobenzene, ethoxybenzene, and anisole as the solvent, and Fig. 2b present the luminance and voltage transients for devices fabricated from such active-material inks stored for 30 days. Table S1 shows that the peak performance has dropped by 20% for the LEC fabricated from the chlorobenzene ink, while the device fabricated from the ethoxybenzene and anisole inks is more robust to long-term ink storage. This difference in long-term stability is attributed to that one component in the multi-solute ink (the n-type host OXD-7) is observed to fall out and crystallize in the chlorobenzene solvent, and that this tendency is inhibited with ethoxybenzene as the solvent, and absent with anisole as the solvent. This suggests that the HSPs of OXD-7 are closer to anisole than chlorobenzene.

We finally evaluated the merits of the three active-material inks for the fabrication of flexible LEC devices by bar coating, which is a much more scalable fabrication method than spin coating. We sequentially bar coated flexible ITO-coated poly(ethylene terephthalate) (PET) substrates with the poly(3,4-ethylenedioxythiophene):poly(styrene sulfonate) ink and the active-material ink. Since a successful deposition by bar coating depends on the reflow of the distributed ink on the substrate before drying for the formation of a uniform film, it is critical to appropriately tune the ink viscosity for the required reflow, the ink surface tension for an appropriate wetting, and the ink boiling point for the desired drying time. Table 2 shows that the surface tension and the boiling point are fairly similar for the three investigated solvents, whereas the larger differences in solvent viscosity are effectively compensated for by the high viscosity introduced by the solute (primarily the high molecular weight host polymer PVK); see "Methods" section for details. Figure S3 shows that uniform thin films can be bar coated with all three active-material inks, and that these bar-coated films can be employed as the active material in LEC devices that deliver bright green and uniform light emission. Figure 2c and Table S2 provide quantitative information on the performance of these bar-coated LECs, and it is notable that the performance is very good and close to that of the spin-coated LEC. The slightly lower luminance and higher drive voltage of the bar-coated LECs in

comparison to the spin-coated LECs can be attributed to that the thickness of the active material of the former is slightly larger than the optimum value of 120 nm. Most importantly, our results demonstrate that both anisole and ethoxybenzene can successfully replace the non-sustainable chlorobenzene for the solvent in a multi-solute LEC ink for both spin-coating and bar-coating fabrication, and that the new sustainable LEC ink can be utilized for an environmentally friendly and safe fabrication of bright, efficient and low-cost LEC devices.

## Discussion

In summary, we introduce a tool for the rational and facile identification of functional and sustainable solvents for the field of printed electronics. Specifically, the tool orders a large number of solvents in accordance to their Hansen solubility and relevant physical parameters, as well as to descriptors and ranking values regarding health, environmental impact, safety, and overall sustainability. We demonstrate the functionality of the tool in a case study on a high-performance light-emitting electrochemical cell, where the currently employed non-sustainable ink solvent was successfully replaced by two alternatives that are more benign. It is our hope that this open-access green-solvent selection tool (freely available at www.opeg-umu.se/green-solvent-tool) will contribute to that dangerous, toxic, and non-sustainable solvents will be efficiently replaced in both research laboratories as well as in larger industrial settings.

## Methods

**Ink preparation.** Master inks were prepared by separately dissolving the solutes poly (9-vinylcarbazole) (PVK, $M_w$ $1.1 \times 10^6$ g mol$^{-1}$, Sigma-Aldrich), 1,3-bis[2-(4-tert-butylphenyl)-1,3,4-oxadiazo-5-yl]benzene (OXD-7, Lumtec), tris[2-(5-substituent-phenyl)-pyridinato]iridium(III) (Ir(R-ppy)$_3$, Merck) and tetrahexylammonium tetrafluoroborate (THABF$_4$, Sigma-Aldrich) in either chlorobenzene (anhydrous, Sigma-Aldrich), ethoxybenzene (Sigma-Aldrich) or anisole (anhydrous, Sigma-Aldrich) under stirring on a hot plate kept at 70 °C for 5 h. The master inks were blended together in a mass ratio of PVK:OXD-7:Ir(R-ppy)$_3$:THABF$_4$ = 32.3:32.3:29.0:6.4 and a total solute concentration as described below for each solvent. The resulting active-material ink was stirred on the hot plate at 70 °C for at least 1 h before further processing. The ink preparation was performed in a N$_2$-filled glovebox ([O$_2$], [H$_2$O] < 1 ppm).

**Device fabrication and characterization.** For the spin-coated LECs, indium-tin-oxide (ITO) coated glass substrates (20 Ω sq$^{-1}$, Thin Film Devices) were carefully cleaned by subsequent sonication in detergent (Extran MA01, Merck), deionized water, acetone and isopropanol followed by drying at 120 °C. Prior to film deposition the substrates were exposed to UV-ozone (model 42–220, Jelight) for 10 min. The poly(3,4-ethylenedioxythiophene):poly(styrene sulfonate) (PEDOT:PSS, Clevios P VP AI 4083, Heraeus) ink was spin coated on top of the ITO at 4000 rpm for 60 s, and dried on a hot plate at 120 °C for 30 min. The thickness of the dry PEDOT:PSS film

was 40 nm. The active-material ink was spin coated onto the PEDOT:PSS, and thereafter dried on a hotplate at 70 °C for 2 h. For the attainment of a 120 nm thick dry active-material film, the following deposition parameters were employed: chlorobenzene: solute concentration: 30 g l$^{-1}$, spin speed: 2000 rpm; ethoxybenzene: 46.5 g l$^{-1}$, 3000 rpm; anisole: 40 g l$^{-1}$, 2000 rpm. On top of the active material, a 100 nm thick Al electrode was deposited by thermal evaporation ($p < 5 \times 10^{-6}$ mbar, Leybold). The $1.5 \times 8.5$ mm$^2$ emission area was defined by the overlap between the ITO anode and the Al cathode. The film thickness was measured with a stylus profilometer (DektakXT, Bruker). The surface uniformity of the spin-coated films was measured with atomic force microscopy (AFM, MultiMode SPM microscope, equipped with a Nanoscope IV Controller, Veeco Metrology) in tapping mode under ambient conditions.

For the bar-coated LECs, ITO (100 Ω sq$^{-1}$) coated PET substrates (thickness = 180 μm) were used. The ITO film was patterned with photolithography, and the patterned PET/ITO substrate was exposed to UV-ozone (model 42–220, Jelight) for 10 min to render its surface hydrophilic. A PEDOT:PSS ink (Clevios P VP AI 4083, Heraeus) was bar coated (AB3000, TQC) on the PET/ITO substrate using a Mayer rod with 0.08 mm wire size and a coating speed of 20 mm s$^{-1}$. The PEDOT:PSS coated substrates were directly and gently transferred onto a hot plate at 120 °C and dried for 4 min. The dry thickness of the bar-coated PEDOT:PSS film was 50–60 nm. The active-material ink (solute concentration = 35 g l$^{-1}$) was bar coated onto the PEDOT:PSS layer, using a Mayer rod with 0.08 mm wire size and a coating speed of 20 mm s$^{-1}$, at room temperature. The coated substrates were directly and gently transferred to a hot plate at 120 °C and dried for 4 min. The dry thickness of the active-material film was 160–220 nm for the chlorobenzene ink, 160–170 nm for the ethoxybenzene ink, and 160–190 nm for the anisole ink. On top of the active material, a 100 nm thick Al electrode was deposited by thermal evaporation. The $2 \times 2$ mm$^2$ emission area was defined by the overlap between the ITO anode and the Al cathode.

The non-encapsulated LEC devices were characterized in an N$_2$-filled glovebox ([O$_2$], [H$_2$O] < 1 ppm). The devices were driven by a constant current density of $j = 77$ A m$^{-2}$ with the voltage compliance set to 20 or 21 V. For the spin-coated LECs, the luminance was measured with a calibrated photodiode (S9219-01, Hamamatsu Photonics) and the voltage was measured with a source measure unit (U2722A, Agilent). For the bar-coated LECs, the luminance and voltage were measured with an OLED lifetime tester (M6000 PMX, McScience). The electroluminescence spectrum was measured with a calibrated spectrometer (USB2000+, Ocean Optics).

**Statistics and reproducibility**. At least six LEC devices were fabricated and characterized for each ink solvent and deposition technique and the LEC data in Fig. 2, Table S1, and Table S2 represent the typical measured performance.

## Data availability
The LEC datasets presented in the current study are available in the Figshare repository via the link: https://doi.org/10.6084/m9.figshare.14786250[42].

## Code availability
The green-solvent selection tool is written in Python, using the Dash framework for web applications, as developed by Plotly[43]. The tool comprises solvent sustainability data and relevant physical properties obtained from References[25,44,45]. The tool features an interactive web interface, which can be accessed free of charge at: www.opeg-umu.se/green-solvent-tool. The source code is available at Github via the link: https://github.com/jrafolsr/opeg-green-solvent/releases and through the Zenodo repository via the link: https://doi.org/10.5281/zenodo.495839546.

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

## Acknowledgements

The authors thank Dr. Helen Sneddon for valuable input and acknowledge financial support from the Swedish Energy Agency, the Swedish Research Council, Kempestiftelserna, the Swedish Foundation for Strategic Research, Stiftelsen Olle Engkvist Byggmästare, and Bertil & Britt Svenssons stiftelse för belysningsteknik.

## Author contributions

C.L., P.L., and L.E. conceptualized the idea. S.T., P.L., C.L., E.M.L., J.W., and A.S. performed the experimental work and the data analysis. J.R.R. and C.L. designed and coded the green solvent tool. L.E., C.L., and P.L. wrote the paper. All authors reviewed the paper.

## Funding

## Competing interests

The authors declare no competing interests.
