## [Peer Review File · Nature Communications]

REVIEWER COMMENTS

Reviewer #1 (Remarks to the Author):

I have previously received this manuscript for review from another journal. After carefully having read the present work, I have seen that it presents minor differences with the previous version. Below you will find my comments to this manuscript and my recommendation for major revisions.

Edman et al present in detail a tool that helps to identify ecofriendly (i.e. green) alternatives to environmentally hazardous solvents commonly used to dissolve functional materials. The selection is made based on the similarity of the Hansen solubility parameters (HSP) of the green solvent alternative to those of the hazardous solvent. The ecofriendliness of the greener alternatives is ranked via a composite score value (G). The manuscript describes in detail the origin of components that are used to calculate it. The manuscript describes the rationale and procedure to develop the solvent identification tool in a comprehensive, consistent and accurate manner. I consider the described procedure and the provided tool are useful to identify greener alternatives to commonly used solvents, however, in its current form the tool is of limited use for the field of printed electronics (PE).

1) As discussed by the authors the Ink formulation process is crucial for PE as it should help yield devices with and optimized performance. The solubility of a material in a solvent is the first condition for a material to be processed by printing technology. Nevertheless, an ink formulation takes into consideration parameters such as viscosity, vapor pressure or surface tension. Therefore, the suggestion of alternative solvents solely on the basis of HSP is incomplete. For example, Ethanol can be suggested as a very good option, however due it is not typically used in PE as it promotes fast drying which usually results in undesirable film topography. In my opinion, this should have been discussed to a greater extent in the manuscript.

2) The specific requirements of the ink formulation would also be determined by the printing technique. Therefore, it would have been important to present LEC devices processed by printing or coating technique instead of utilizing spin coating. This technique is not relevant for the PE field. It is very common that lab-scale processes based on spin-coating require an ink formulation step to be transferred to an up scalable technique. Utilizing a scalable technique would have supported the authors' claim that the presented tool will have an impact in the PE field.

3) In the experimental section, the authors declare that they needed different concentrations of solute in the different solvents to achieve the same thickness at constant rpms. This infers that either viscosity, or surface tension were changed and that the new green formulation did not provide a 1:1 replacement. This is not unexpected but should have been discussed.

4) The authors presented a comparison of the LEC performance when casted from the different inks showing very good results. However, after ink storage the results of the ethoxybenzene formulation are not presented. Is there any relevant reason for it?

5) It would have been interesting to see a characterization of the deposited layers by microscopy, or

investigate the homogeneity of the printed pixels to check if the solvent had any influence on these properties.

In order to obtain the best results, the use of co-solvent systems is sometimes required. Such approach considering green solvents was investigated to some extent in <https://doi.org/10.1002/adfm.201301509>. Here, the authors also determined green-solvent alternatives on the basis of HSP and compared device performance after depositing via blade coating taking into account the vapor pressure of the new formulation. This reference has now been included in the manuscript however no discussion was provided.

In summary, the solvent selection tool presented in this manuscript has been carefully developed and could be used as a starting point for an ink formulation process based on green solvents. However, I do not consider that, in its current form, it provides further insights into the PE field that would grant its publication in Nature Communications.

Reviewer #2 (Remarks to the Author):

Review of: "A tool for identifying green solvents for printed electronics"

The paper is presented well and provides an overview of a new useful on-line tool for the identification of alternative more green solvent options. A specific case study is presented for the usage of the new tool for the replacement of solvents for the fabrication of light-emitting electrochemical cell.

Some minor corrections are noted below.

Change: "Thus, it is easy to motivate why alternative "green" solvents, which deliver an ink performance on par with currently used non-sustainable solvents, should be identified."

To: "Thus, it is easy to find motivation for the selection of alternative "green" solvents, which deliver an ink performance on par with currently used non-sustainable solvents."

Change: "... but informed parties can most probably agree that a qualified green solvent should exhibit a combination of low health hazard, high safety, and small environmental impact during its entire life cycle."

To:

"... but informed parties can most probably agree that a qualified green solvent should exhibit a combination of a low health hazard, a high level of safety, and a small environmental impact during its entire life cycle."

The text related to the table and figures should ideally come before the corresponding table and figure. It appears mostly after the corresponding table and figure in the paper.

Reviewer 1

The Reviewer states that our “manuscript describes the rationale and procedure to develop the solvent identification tool in a comprehensive, consistent and accurate manner” and that “the described procedure and the provided tool are useful to identify greener alternatives to commonly used solvents.” The Reviewer also presents a number of areas for improvement as detailed below:

1) As discussed by the authors the Ink formulation process is crucial for PE as it should help yield devices with and optimized performance. The solubility of a material in a solvent is the first condition for a material to be processed by printing technology. Nevertheless, an ink formulation takes into consideration parameters such as viscosity, vapor pressure or surface tension. Therefore, the suggestion of alternative solvents solely on the basis of HSP is incomplete. For example, Ethanol can be suggested as a very good option, however due it is not typically used in PE as it promotes fast drying which usually results in undesirable film topography. In my opinion, this should have been discussed to a greater extent in the manuscript.

Our response: We first wish to thank the Reviewer for highly appreciated insightful and constructive input. We completely agree with the Reviewer that other parameters than the solubility parameters (in our manuscript the HSPs) also play a critical role in determining the merit of an ink solvent. In line with the Reviewer’s suggestion, we have therefore added information on the viscosity and surface tension (information on the vapor pressure was already part of the tool through the boiling point) for the different solvents, and also further developed the tool so that it is possible to define an allowed range for these three properties for a functional solvent. We have further included a discussion in the text on page 9 (lines 151-164) to highlight that these three parameters should be considered during the selection of a functional replacement solvent, and also discussed these specific requirements as regards to the case study on a number of locations (marked with yellow background) on pages 13-15.

2) The specific requirements of the ink formulation would also be determined by the printing technique. Therefore, it would have been important to present LEC devices processed by printing or coating technique instead of utilizing spin coating. This technique is not relevant for the PE field. It is very common that lab-scale processes based on spin-coating require an ink formulation step to be transferred to an up scalable technique. Utilizing a scalable technique would have supported the authors’ claim that the presented tool will have an impact in the PE field.

Our response: The Reviewer brings forward very valid points in that each printing/coating technique requires its own ink formulation, and that spin coating is not upscalable (albeit commonly used in PE research). We have therefore followed the Reviewer’s advice, and after quite a lot of effort managed to successfully fabricate high-performance LEC devices with upscalable bar coating. These new results are presented in new Figures 2c and S3 and Table S2 and in the accompanying text. This achievement required a reformulation of the active-material inks, as discussed in detail on pages 14 and 16-17. However, the main result of the case study remains, *viz.* that it is possible to utilize benign ethoxybenzene and anisole instead of less-appealing chlorobenzene for the ink solvent for the scalable fabrication of high-performance LEC devices.

3) In the experimental section, the authors declare that they needed different concentrations of solute in the different solvents to achieve the same thickness at constant rpms. This infers that either viscosity, or surface tension were changed and that the new green formulation did not provide a 1:1 replacement. This is not unexpected but should have been discussed.

Our response: The Reviewer brings forward a relevant point, and we have included a section of text that addresses this issue on page 13 of the revised manuscript.

4) The authors presented a comparison of the LEC performance when casted from the different inks showing very good results. However, after ink storage the results of the ethoxybenzene formulation are not presented. Is there any relevant reason for it?

Our response: The reason for the lack of ink-storage data for ethoxybenzene was that we had chosen to focus on one of the green replacement solvents for this more time-consuming study. However, since the new bar-coating experiments turned out to be very time consuming, we decided to also complement the ink-storage study with the missing data for ethoxybenzene. These new data are now presented in the revised Figure 2b on page 12.

5) *It would have been interesting to see a characterization of the deposited layers by microscopy, or investigate the homogeneity of the printed pixels to check if the solvent had any influence on these properties.*

Our reponse: We have followed the Reviewer's advice and performed studies on the microscopic surface morphology with AFM and the larger-scale film uniformity by photoluminescence mapping. These new data are presented in new Figure S1, which reveal that the selection of ink solvent had essentially no measureable influence on the microscopic and macroscopic structure.

6) *In order to obtain the best results, the use of co-solvent systems is sometimes required. Such approach considering green solvents was investigated to some extent in <https://doi.org/10.1002/adfm.201301509>. Here, the authors also determined green-solvent alternatives on the basis of HSP and compared device performance after depositing via blade coating taking into account the vapor pressure of the new formulation. This reference has now been included in the manuscript however no discussion was provided.*

Our response: The Reviewer points to an important opportunity for the appropriate formulation of advanced inks and/or for the achievement of complex dry-film morphologies, and we have therefore included a discussion on the merit of the co-solvent approach in the context of ink functionality and complex film morphology on page 9.

Reviewer 2

The Reviewer states that our "paper is presented well and provides an overview of a new useful on-line tool for the identification of alternative more green solvent options." He/she also provides a number of suggestions for grammatical improvement as noted below.

1. *Change: "Thus, it is easy to motivate why alternative "green" solvents, which deliver an ink performance on par with currently used non-sustainable solvents, should be identified." to: "Thus, it is easy to find motivation for the selection of alternative "green" solvents, which deliver an ink performance on par with currently used non-sustainable solvents."*

Change: "... but informed parties can most probably agree that a qualified green solvent should exhibit a combination of low health hazard, high safety, and small environmental impact during its entire life cycle." to: "... but informed parties can most probably agree that a qualified green solvent should exhibit a combination of a low health hazard, a high level of safety, and a small environmental impact during its entire life cycle."

Our response: With the caveat that we are not native English speakers, our feeling is that the original formulation better captures the message that we wish to convey. However, if the Editor is of a different opinion then we are definitely ready to reconsider.

2. *The text related to the table and figures should ideally come before the corresponding table and figure. It appears mostly after the corresponding table and figure in the paper.*

Our response: We are confident that the Editorial office will help us position the tables and figures at the best possible position, and are def

REVIEWERS' COMMENTS

Reviewer #1 (Remarks to the Author):

Edman et al have completely addressed and clarified the points i raised on their previous version of the manuscript. In particular, i appreciate the time the authors took to perform new experiments utilizing bar coating and the careful revision of the discussion section.

I do not have further comments to the manuscript and recommend it for publication in Nature Communications.